# TREM2 Alleviates Neuroinflammation by Maintaining Cellular Metabolic Homeostasis and Mitophagy Activity During Early Inflammation

**DOI:** 10.3390/diseases13020060

**Published:** 2025-02-16

**Authors:** Lingfeng Hu, Jie Liu, Jie Peng, Xiao Li, Zhangqiong Huang, Caixing Zhang, Shengtao Fan

**Affiliations:** Institute of Medical Biology, Chinese Academy of Medicine Sciences & Peking Union Medical College, Kunming 650108, China; 13187715267@163.com (L.H.); 13518303106@163.com (J.L.); 15332644336@163.com (J.P.); s2023018001@pumc.edu.cn (X.L.); hzhq@imbcams.com.cn (Z.H.); 2675740248@imbcams.com.cn (C.Z.)

**Keywords:** TREM2, neuroinflammation, mitochondrial, mitophagy, microglia

## Abstract

Aims: Inflammation is a pivotal characteristic of neurodegenerative diseases. The triggering receptor expressed on the myeloid cells 2 (TREM2) gene has previously been shown to suppress inflammation by directly inhibiting inflammation-related pathways. Mitochondrial dysfunction has recently emerged as another critical pathological manifestation of neurodegenerative diseases. Although TREM2 is involved in the regulation of cellular energy metabolism and mitochondrial autophagy, its role in the relationship between inflammation and mitochondrial autophagy remains unclear. Methods: In this study, we generated TREM2-overexpressing BV-2 cells and established a neuroinflammatory model with LPS. We compared these cells with wild-type cells in terms of inflammation, metabolism, autophagy, and mitochondria using methods such as RT–qPCR, Western blotting, immunocytochemistry, transmission electron microscopy, and flow cytometry. Results: Microglia overexpressing TREM2 exhibited increased resistance to inflammation. Additionally, these cells inhibited the metabolic reprogramming that occurs early in LPS-induced inflammation, reduced ROS release, mitigated mitochondrial damage, maintained a certain level of autophagic activity, and cleared damaged mitochondria. Consequently, they alleviated the inflammation caused by the mitochondrial barrier. Conclusions: ur results suggest that TREM2 can alleviate inflammation by maintaining cellular metabolic homeostasis and mitochondrial autophagy activity.

## 1. Introduction

The triggering receptor expressed on the myeloid cells 2 (Trem2) gene has significant research value in the field of neuroinflammation. Trem2 encodes myeloid cell triggering receptor 2, which is expressed primarily by microglia in the central nervous system and plays a crucial role in regulating microglial survival, activation, and inflammatory responses [1]. In recent years, with in-depth studies of neurodegenerative diseases, especially Alzheimer’s disease (AD), the prominent role of the Trem2 gene has become increasingly apparent, including its functions in suppressing neuroinflammation and regulating metabolism [2,3,4,5,6].

The early stages of AD are accompanied by neuroinflammation, metabolic alterations, and mitochondrial dysfunction [7,8]. In the brains of AD patients, vulnerable neurons exhibit impaired mitochondrial autophagy and abnormal mitochondria accumulation within the cytoplasm, indicating that impaired mitochondrial autophagy may be one of the pathological hallmarks of AD [9]. The amyloid-β (Aβ) protein and hyperphosphorylated tau protein interact with other mitochondrial proteins (such as Drp1, VDAC, and CypD), leading to excessive mitochondrial fragmentation, mitochondrial dynamics disruption, and impaired mitochondrial autophagy [10]. These events ultimately result in multiple types of neuronal dysfunction, contributing to the progression of inflammation and a decline in cognitive function [11]. Previous studies have revealed varying degrees of mitochondrial dysfunction in AD patients, as well as in AD model mice, specifically the 3xTg and 5xFAD strains [12], including damage, swelling, excessive accumulation, and the release of mtDNA (mitochondrial DNA). Notably, mtDNA can activate the cGAS-STING pathway, TLR receptors, and inflammasome formation, serving as crucial factors in the progression of AD [13,14]. To maintain the homeostasis of mitochondria and energy metabolism and prevent the accumulation of damaged mitochondria within cells, cells primarily rely on the mitochondrial autophagy pathway [15]. This process is a form of cellular autophagy that selectively eliminates damaged mitochondria to maintain the stability of mitochondria and energy metabolism, thereby preventing the accumulation of pathological mitochondria within the cell [12].

Trem2 has previously been shown to regulate NF-κB-related pathways to control inflammation [16]. Furthermore, downregulated Trem2 expression has been observed in AD animal models and microglial inflammation models [4,17]. Additionally, the absence or mutation of Trem2 can lead to metabolic alterations in microglia under amyloid protein stimulation [5,18]. This manifests as a shift from oxidative phosphorylation to aerobic glycolysis (Warburg effect), accompanied by the activation of the AKT–mTOR pathway and a decrease in autophagy [19,20]. On the basis of the aforementioned findings, we hypothesize that regulating the NF-κB pathway is not the sole mechanism by which Trem2 counters neuroinflammation. Trem2 may also regulate cellular energy metabolism, thereby influencing the level of mitochondrial autophagy. By clearing damaged mitochondria and reducing the release of mtDNA-induced inflammation, Trem2 ultimately exerts an anti-inflammatory effect through this pathway.

To uncover potential additional mechanisms of Trem2 in neurodegenerative diseases, particularly neuroinflammation, we employed wild-type BV-2 microglia and Trem2-overexpressing BV-2 cells as models. By stimulating cells with lipopolysaccharide (LPS) to establish an early neuroinflammatory cell model [21], we compared the performance of the two cell types in terms of inflammation, metabolism, and mitochondrial autophagy. Our findings revealed that the Trem2-overexpressing cell model maintained a certain level of autophagic activity, preserved mitochondrial health, inhibited inflammation-induced early metabolic shifts, and thus resisted inflammation.

## 2. Materials and Methods

### 2.1. Reagents

Dulbecco’s modified Eagle’s medium (DMEM), penicillin, and streptomycin were purchased from Gibco BRL (Grand Island, NY, USA). Lipopolysaccharide (LPS, L6529) and 0.25% trypsin were obtained from Sigma Chemical Co. (St. Louis, MO, USA). Fetal bovine serum (FBS) was purchased from ExCell Bio (Suzhou, China). A Takara MiniBEST Universal RNA Extraction Kit, Takara MiniBEST Universal Genomic DNA Extraction Kit Ver 5.0, PrimeScrip^TM^ RT Reagent Kit with gDNA Eraser, and TB Green Premix EX Taq^TM^ II were purchased from Takara Biotechnology (Dalian, China). H2DCFDA (HY-D0940), Phosphatase Inhibitor Cocktail III (100× in DMSO), and Protease Inhibitor Cocktail (EDTA-Free, 100× in DMSO) were obtained from MedChemExpress (MCE, Monmouth Junction, NJ, USA). PK Mito Red (PKMR-2) was obtained from Genvivo Biotech (Nanjing, China). MitoBright LT Red (MT11) and MitoBright ROS Deep Red-Mitochondrial Superoxide Detection (MT16) were purchased from Dojindo Laboratories (Kumamoto, Japan). Seahorse XF DMEM, pH 7.4, Seahorse XF 1.0 M glucose solution, Seahorse XF 100 mM pyruvate solution, Seahorse XF 200 mM glutamine solution, Seahorse XF Calibrant, and a Seahorse XFp Glycolytic Rate Assay Kit (103346-100) were purchased from Agilent Technologies (Santa Clara, CA, USA).

The antibodies used in this research included anti-cyclooxygenase 2 (1:1000; Zenbio, Chengdu, China, R23971), anti-SQSTM1/p62 (1:1000; Zenbio, Chengdu, China, 382862), anti-β-actin (1:10,000; Zenbio, Chengdu, China, 380624), anti-LC3 (1:1000; CST, Danvers, MA, USA, 12741), anti-LC3 (1:200; Proteintech, Wuhan, China, 81004-1-RR), goat anti-rabbit recombinant secondary antibody (H + L) (1:10,000; Proteintech, Wuhan, China, RGAR001), and anti-iNOS (1:1000; Proteintech, Wuhan, China, 18985-1-AP).

### 2.2. Cell Culture

BV-2 cells were acquired from ZQXZBIO (Shanghai, China) and mouse Trem2-overexpressing BV-2 cells (designated T2OE-BV-2) were obtained from EDITGENE (Guangzhou, China). Both cell lines were cultured in DMEM supplemented with 10% FBS at 37 °C in a humidified atmosphere containing 5% CO_2_, all cells used in the experiments were cultured up to the 5th generation post resuscitation before utilization, and none were used beyond the 15th generation in the study.

Before the experiments, BV-2 and T2OE-BV-2 cells were seeded into 6-well plates at a density of 5 × 10^5^ cells per well, and these cells were cultured for 12 h. Stock solutions of LPS, prepared by diluting LPS in PBS to a concentration of 100 μg/mL, were further diluted in cell culture medium to the required working concentrations of 1000 ng/mL and 100 ng/mL. These LPS concentrations were chosen on the basis of the findings reported by Cheng et al. [19] and were applied to the cells for 4 h to induce an inflammatory response.

### 2.3. RNA Isolation and Quantitative Real-Time PCR

Total RNA was extracted from both BV-2 and T2OE-BV-2 cells using a Takara MiniBEST Universal RNA Extraction Kit (TaKaRa, Tokyo, Japan). The extracted RNA was then reverse transcribed using a PrimeScript™ RT reagent Kit with gDNA Eraser (TaKaRa, Tokyo, Japan), ensuring the removal of any genomic DNA contamination. Real-time quantitative PCR was conducted on a CFX96 Real-Time PCR system (Bio-Rad, Hercules, CA, USA), with TB Green Premix EX Taq™ II as the fluorescent dye. The specific primers utilized in the reactions were procured from Sangon Biotech (Shanghai, China), and are listed in Table 1. Actin served as the housekeeping gene for normalization purposes. The normalized Ct values were calculated using the comparative ΔΔCt method.

### 2.4. Relative mtDNA Release Analysis

The relative mtDNA content was measured according to a previously described method [22]. The DNA was extracted from cells using a MiniBEST Universal Genomic DNA Extraction Kit Ver 5.0 (Takara, Dalian, China), and the mtDNA content was measured via qPCR. The primers used were procured from Sangon Biotech (Shanghai, China), and are listed in Table 2. The Tert gene served as the housekeeping gene for the normalization of nuclear DNA, and the Dloop1 gene served as the housekeeping gene for the normalization of mtDNA.

### 2.5. Western Blot Analysis

BV-2 and T2OE-BV-2 microglia were subjected to a comprehensive processing protocol as follows. Initially, the microglia were rinsed with phosphate-buffered saline (PBS) and lysed on ice for 15 min in denaturation lysis buffer supplemented with phosphatase inhibitor cocktail III (diluted 100× in DMSO) and an EDTA-free protease inhibitor cocktail (also diluted 100× in DMSO). The lysates were sonicated on ice until clear, indicating complete cellular lysis and efficient protein extraction. The total protein concentration in the lysates was then quantitatively assessed using a BCA protein assay kit (Thermo Fisher Scientific, Waltham, MA, USA). Subsequent SDS–PAGE separation was conducted using gels of various concentrations (7.5%, 10%, and 15%) on the basis of the molecular weights of the target proteins. The resolved proteins were transferred onto a polyvinylidene difluoride (PVDF) membrane, which was blocked with flash blocking buffer (Affinibody, Wuhang, China) for 10 min to prevent nonspecific antibody binding. The membrane was then incubated with primary antibodies specific to the proteins of interest, followed by incubation with horseradish peroxidase (HRP)-conjugated secondary antibodies. Finally, the protein bands were visualized using enhanced chemiluminescence (ECL) reagents (Merck Millipore, Billerica, MA, USA) and a ChemiDoc Imaging System (Bio-Rad, CA, USA).

### 2.6. Mitochondrial Flow Cytometry

BV-2 and T2OE-BV-2 cells were treated with LPS for 4 h. Following this treatment, the cells were washed with PBS and subsequently incubated with DMEM containing either MitoBright LT Red or MitoBright ROS Deep Red (for mitochondrial superoxide detection) for 15 min in a CO_2_ incubator. After staining, the cells were washed twice with PBS and resuspended. Fluorescence intensity was then quantitatively assessed via flow cytometry using equipment from Beckman Coulter (Miami, FL, USA).

### 2.7. ROS Flow Cytometry

Following treatment with LPS, BV-2 and T2OE-BV-2 cells were processed in manner similar to that described above for cells used for mitochondrial flow cytometry. The cells were washed and incubated with H2DCFDA (diluted 1:1000) in DMEM for 30 min in a CO_2_ incubator. Then, the cells were digested with 0.25% trypsin and centrifuged at 300× *g* for 5 min. The cells were then washed, resuspended in DMEM without phenol red, and subjected to flow cytometry analysis using equipment from Beckman Coulter (Brea, CA, USA) to measure the fluorescence intensity.

### 2.8. Immunocytochemistry

BV-2 and T2OE-BV-2 cells were seeded in a 24-well plate at a density of 5 × 10^4^ cells per well and allowed to adhere overnight. The cells were subsequently treated with LPS. After treatment, the cells were fixed with 4% paraformaldehyde for 15 min at room temperature, permeabilized with 0.5% Triton X-100 in PBS for 10 min, and then blocked with 5% BSA for 1 h at room temperature. The cells were washed with PBS and incubated with primary antibodies overnight at 4 °C. Following three washes with PBS, the cells were stained with Alexa Fluor-conjugated secondary antibodies for 1 h at room temperature in the dark. After washing, the cells were mounted with ProLong Gold mounting medium containing DAPI. Images were captured using a Leica Sp8 confocal microscope and analyzed using ImageJ software(version 1.54h 15 December 2023).

### 2.9. Transmission Electron Microscopy (TEM)

After the cells were treated with LPS, they were harvested, resuspended, and fixed in TEM fixative at 4 °C for preservation and transportation. Agarose blocks containing the samples were protected from light and incubated with 1% OsO_4_ in 0.1 M PB (pH 7.4) for 2 h at room temperature. After OsO_4_ was removed, the tissues were rinsed with 0.1 M PB (pH 7.4) 3 times for 15 min each. The samples were subsequently processed through a graded ethanol series, embedded in araldite, and polymerized at 37, 45, and 60 °C for 24 h at each temperature. The blocks were cut into 60–80 nm sections on an ultramicrotome (Leica UC7), and the sections were placed on 150 mesh cuprum grids with Formvar film. The sections were incubated with 2% uranium acetate saturated alcohol solution in the dark for 8 min, rinsed with 70% ethanol 3 times, and then rinsed with ultrapure water 3 times. The sections were then incubated with 2.6% lead citrate without CO_2_ for 8 min and then rinsed with ultrapure water 3 times. After being dried with filter paper, the cuprum grids were placed on a grid board and dried overnight at room temperature. The cuprum grids were observed using a transmission electron microscope (HITACHI, HT7800), and images were captured.

### 2.10. Metabolic Extracellular Flux Analysis

The bioenergetic properties of BV-2 microglia under different conditions were determined using an XFp Seahorse extracellular flux analyzer (Agilent, Santa Clara, CA, USA), which measures real-time changes in the extracellular acidification rate (ECAR) and the oxygen consumption rate (OCR), which are measures of glycolysis and mitochondrial respiration, respectively. BV-2 or T2OE-BV-2 cells were seeded on XFp cell culture plates at approximately 2 × 10^4^ cells per well in the presence or absence of LPS. The probe plates were hydrated overnight. The cells were washed with running buffer (XF assay medium, 10 mM glucose, 1 mM pyruvate sodium, and 2 mM L-glutamine) and the probe plates were calibrated. The real-time glycolytic rate kit contained two drugs, including rotenone and 2-DG (2-Deoxy-D-glucose). Rotenone inhibits mitochondrial complex I, shifting cellular energy production from oxidative phosphorylation to glycolysis, and 2-DG competitively inhibits hexokinase, thereby blocking glycolysis. The drugs were added into the injection port in accordance with the instruction manual. After completion of the measurements, the number of cells within the plate was determined via Hoechst 33,342 staining, and the cells were counted using Cytation 5 (BioTek, Biotek Winooski, VT, USA). The data were normalized to the average number of cells.

### 2.11. Statistical Analysis

All values are expressed as means ± SDs. Statistical analysis was carried out using GraphPad Prism 10 software for Windows. For three or more groups, the data were analyzed via one-way ANOVA, followed by Tukey’s HSD test. For two experimental groups, the data were analyzed using Student’s *t* test. A value of *p* < 0.05 was considered significant.

## 3. Results

### 3.1. LPS-Induced Inflammation Is Regulated by Trem2

To investigate the effect of the LPS dose on microglia-induced inflammation, we treated wild-type BV-2 cells with different concentrations of LPS (1000 ng/mL and 100 ng/mL) for 4 h. We observed an increase in the mRNA expression of inflammation-related factors at both LPS concentrations, with the high-concentration group (1000 ng/mL) exhibiting increased expression levels of inflammatory factors (Figure 1A). Moreover, the expression of the TREM2 gene decreased, and the greater the degree of inflammation was, the greater the degree of the downregulation of TREM2 gene expression (Appendix A). LPS at a high dose effectively induced inflammation and led to the downregulation of TREM2 expression. To validate these findings, we generated T2OE–BV-2 cells and treated them with 1000 ng/mL LPS for 4 h. The mRNA expression levels of TNF-α, IL-1β, and IL-6 were significantly lower in T2OE–BV-2 cells than in wild-type BV-2 cells (Figure 1B). To further confirm these results, we measured the expression levels of inflammation-related proteins. WB results revealed that LPS induced microglial inflammation, and that the expression levels of COX-2 and iNOS were significantly lower in the TREM2-overexpressing cell group than in the wild-type cell group (Figure 1C,D). These results collectively indicate that LPS can effectively induce inflammation, and that LPS-induced inflammation is regulated by TREM2 in mouse microglia. The overexpression of the TREM2 gene can suppress inflammation.

### 3.2. Trem2 Counteracts the Metabolic Transition of Microglia in the Early Stages of Inflammation

The upregulation of inflammatory factor expression in immune cells is often accompanied by alterations in cellular energy metabolism [23]. To determine whether Trem2 inhibits inflammation by counteracting such metabolic shifts, we examined the key enzymes and transporters involved in cellular glycolytic metabolism. Ldha promotes lactate production, sustaining glycolysis; HK2 initiates glycolysis, catalyzing glucose phosphorylation; and GLUT1 facilitates glucose uptake, supporting cellular metabolism. RT–qPCR analysis was performed using wild-type and TREM2-overexpressing (T2OE) BV-2 cells at 2 and 4 h post stimulation with LPS. We found that, in wild-type cells, the expression levels of GLUT1 and HK2, which are enzymes related to early glycolytic metabolism, increased after LPS treatment, with significant differences compared to the expression levels of GLUT1 and HK2 in the control cells at the 4 h mark (Figure 2A). These findings indicate an increase in glycolysis in wild-type cells at this time point. However, there were no significant differences in the expression of glycolytic enzymes in T2OE–BV-2 cells (Figure 2B). Additionally, a decrease in Ldha expression was observed in both cell types (Figure 2A,B), suggesting that LPS can induce metabolic reprogramming in microglia, accompanied by lactate accumulation.

We utilized a Seahorse real-time energy metabolism analyzer to measure the real-time glycolysis rates of both cell types following LPS stimulation. Within the first 20 min of measurement, the basal glycolysis rate of treated wild-type cells was the highest. When the electron transport chain was blocked (by the addition of rotenone), the maximum glycolysis rate of unstimulated TREM2-overexpressing (T2OE) BV-2 cells was only slightly greater than that of treated T2OE cells, and significantly lower than that of wild-type cells. Furthermore, the maximum glycolysis rate of wild-type cells after LPS treatment was significantly greater than that of the unstimulated group (Figure 3A). The results revealed that the average proton efflux rate (PER) of wild-type cells increased significantly after treatment, whereas there was no significant increase in the average PER of treated TREM2-overexpressing (T2OE) BV-2 cells. The ECAR is a key indicator of glycolytic metabolism, and the OCR reflects oxidative phosphorylation. Wild-type BV-2 cells presented increased glycolysis and increased oxygen consumption after treatment with LPS, leading to an increased energy demand (Figure 3B,C). This suggests that metabolic reprogramming occurred, i.e., a shift toward aerobic glycolysis (Warburg effect). In contrast, there were no significant differences in the ECAR or OCR in T2OE–BV-2 cells after LPS stimulation, indicating that the overexpression of the TREM2 gene can prevent metabolic shifts in microglia following treatment.

### 3.3. Mitochondrial Damage Occurs in Microglia After Stimulation with LPS

Inflammation often leads to an increase in intracellular reactive oxygen species (ROS), which can damage mitochondria when present in excess. By measuring the intracellular ROS levels in cells after LPS treatment, we found that the ROS content increased in wild-type BV-2 cells (Figure 4A), while there was a slight increase in ROS in T2OE–BV-2 cells; however, the difference was not statistically significant. Additionally, in the unstimulated state, the ROS content was greater in the wild-type cells than in the T2OE–BV-2 cells. To determine whether these ROS were generated within the mitochondria, we used the mitochondrial ROS probe MITOSOX. The trend observed with MITOSOX was generally consistent with that of intracellular ROS (Figure 4B). Specifically, mitochondrial ROS levels increased in treated wild-type BV-2 cells, whereas there was only a slight increase in mitochondrial ROS in T2OE–BV-2 cells; the difference was not statistically significant. These findings suggest that the total cellular ROS and ROS released from mitochondria increased after LPS stimulation, but that the damage caused by ROS in TREM2-overexpressing cells was significantly lower than that in wild-type cells. Therefore, the TREM2 gene may be involved in the regulation of cellular energy metabolism, thereby reducing the degree of mitochondrial damage caused by excessive ROS release.

To confirm the status of mitochondrial damage, we measured the release of intracellular mtDNA (mitochondrial DNA molecules that are released into the cytoplasm when mitochondria are damaged). The results revealed that the release of mtDNA in LPS-treated, wild-type cells increased, and that there was no significant increase in the release of mtDNA in the T2OE–BV-2 cells (Appendix A). These findings indicate that mitochondrial damage was more severe in wild-type cells after treatment with inflammatory stimuli.

When a large number of mitochondria are damaged, they cannot be rapidly cleared, leading to mitochondrial accumulation. We labeled and detected intracellular mitochondria via flow cytometry. Our findings revealed that both cell types exhibited mitochondrial accumulation under inflammatory conditions. However, this accumulation was more severe in wild-type cells than in TREM2-overexpressing cells (Figure 4C).

To visually observe the damage to mitochondria, we performed transmission electron microscopy using stimulated wild-type BV-2 cells and T2OE–BV-2 cells. Damaged and swollen mitochondria were observed in the treated wild-type cells (Figure 4D), whereas the mitochondria in the T2OE–BV-2 cells maintained a normal morphology with clear cristae (Figure 4E).

### 3.4. Under Inflammatory Conditions, Microglial Autophagy Is Impaired

Given that the degree of mitochondrial damage caused by inflammation is less severe in TREM2-overexpressing microglia than in wild-type cells, and that there is less mitochondrial accumulation (Figure 4C,D), we hypothesized that the greater degree of mitochondrial autophagy in TREM2-overexpressing cells removes damaged mitochondria in a timely manner to prevent the further exacerbation of cellular inflammation. To test this hypothesis, we first examined the expression of autophagy-related genes. The RT–qPCR results revealed that, compared with wild-type cells, T2OE–BV-2 cells presented a lower degree of downregulated ATG5 and ATG7 (core proteins for autophagosome formation and maturation) expression after LPS treatment (Figure 5A). To analyze autophagic flux, we measured the expression levels of LC3 and SQSTM1/p62. We found that in both cell types, p62 accumulated after stimulation (Figure 5B). However, compared with T2OE–BV-2 cells, wild-type cells tended to convert LC3-II back to LC3-I after LPS treatment, whereas T2OE–BV-2 cells remained largely unchanged (Figure 5C). Additionally, we used immunofluorescence to determine the intracellular distribution of LC3 (Figure 6A), and found that LC3 expression decreased in wild-type cells after treatment, with a weakened fluorescence intensity compared to that in normal cells, a finding that was consistent with the WB results (Figure 6B). These findings suggest that the overexpression of TREM2 can increase resistance to the changes in autophagic flux caused by inflammation.

When assessing internal cellular structures via TEM, we found that autophagosomes were difficult to locate in wild-type cells after LPS treatment (Figure 7A). However, autophagosomes were more easily identifiable in LPS-treated T2OE–BV-2 cells (Figure 7B). These findings indicate that LPS-stimulated T2OE–BV-2 cells maintained a certain level of autophagic activity, thereby clearing damaged mitochondria.

## 4. Discussion

Our experimental results revealed that in addition to inhibiting the NF-κB pathway [24], TREM2 enhanced the ability of cells to resist early metabolic shifts induced by inflammation, and to maintain mitochondrial autophagic activity, thereby mitigating mitochondrial damage caused by inflammation-promoted ROS release. Furthermore, TREM2 promoted the ability of cells to sustain a certain level of autophagic activity, alleviating further inflammation exacerbation due to mitochondrial fragmentation and mtDNA release. Microglia are the most crucial players in neuroinflammation [25]. BV-2 cells, which are mouse microglia, exhibit a significant decrease in TREM2 expression upon LPS stimulation, an effect that is negatively correlated with inflammation severity. Previous studies have also indicated that TREM2 is a key gene in neuroinflammation [26]. Therefore, TREM2 serves as an important regulatory factor in neuroinflammation. We constructed TREM2-overexpressing BV-2 cells using a lentivirus and used LPS to establish a microglial neuroinflammation model.

Early neuroinflammation is often accompanied by the metabolic reprogramming of immune cells, which is characterized by a shift from oxidative phosphorylation to aerobic glycolysis. This shift occurs due to the increased energy demand of cells during the initial stages of inflammation, leading to accelerated glycolysis rates [18]. Therefore, we examined the expression levels of inflammatory cytokines four hours after LPS treatment and found that the expression of proinflammatory cytokines such as IL-6, IL-1β, and TNF-α was significantly increased at this time point. However, the expression levels of these cytokines were lower in TREM2-overexpressing cells than in wild-type cells, suggesting that increased TREM2 expression has a significant anti-inflammatory effect. Simultaneously, we measured the expression levels of glycolysis-related enzymes such as Ldha and HK2, and glucose transporter protein GLUT1, as well as the real-time glycolysis rate and maximum glycolysis capacity of the cells. Before the addition of rotenone, the basal glycolysis rate of unstimulated wild-type cells was lower than that of LPS-stimulated WT–BV-2 cells. This may be due to the fact that TREM2 overexpression suppresses the JAK/STAT axis and its downstream mTOR pathway [27,28], leading to reduced basal glycolysis in the overexpressing cells, whereas the basal glycolysis rate of TREM2-overexpressing microglia essentially remained unchanged after LPS stimulation and was lower than that of wild-type cells. When rotenone was injected to completely inhibit oxidative phosphorylation, the maximum glycolysis rate of the cells was measured. We found that wild-type cells underwent metabolic reprogramming four hours after LPS treatment, with increased glycolysis-related enzyme expression and elevated basal and maximum glycolysis rates. In contrast, compared with untreated TREM2-overexpressing cells, LPS-treated TREM2-overexpressing cells exhibited resistance to this metabolic shift, with no significant increase in glycolytic capacity. This may be a result of the ability of TREM2 to suppress inflammation, leading to a reduced energy demand. The reason for this outcome could be that TREM2 is an upstream regulator of the key metabolic regulatory gene mTOR [29]. Cellular energy metabolism involves mainly glycolysis and oxidative phosphorylation, and the dynamic balance between them maintains cellular homeostasis. The state of mitochondria is important for maintaining homeostasis. After observing enhanced glycolysis in microglia under inflammatory stimulation, we speculate that mitochondrial damage may have occurred, leading to impaired oxidative phosphorylation.

The generation of inflammation often leads to an increase in the levels of intracellular reactive oxygen species (ROS) and other peroxides and superoxides, which can attack cellular structures, exacerbating inflammation or inducing apoptosis [30]. In the results, we observed a phenomenon where intracellular ROS levels were reduced in overexpressing cells. This may be due to TREM2 enhancing the cells’ ability to clear ROS. Studies have shown that TREM2 activates the PI3K/Akt pathway, increasing the expression of SOD (superoxide dismutase) and GSH-Px (glutathione peroxidase) [31]. In addition, the inability of damaged mitochondria to be cleared in a timely manner can also lead to the release of a large amount of ROS into the cell. This could result in overexpressing cells having a stronger capacity to scavenge ROS, thereby exhibiting lower overall ROS levels. Our results revealed that, compared with TREM2-overexpressing cells, WT cells produced more ROS after LPS stimulation, and that pyroptosis may serve as a critical mechanism underlying mitochondrial damage-induced ROS release. LPS triggers GSDMD activation, which subsequently induces pore formation in mitochondrial membranes, ultimately leading to intracellular ROS liberation [32]. To determine whether the trend was consistent between intracellular and mitochondrial ROS, we measured mitochondrial peroxides and found that the results were consistent with the results of the intracellular measurements. Excessive ROS can cause mitochondrial membrane damage, leading to mitochondrial swelling and rupture, in turn leading to the release of mtDNA into the cytoplasm [33]. By extracting intracellular DNA to assess mtDNA release, we found that wild-type cells released more mtDNA into the cytoplasm, and that the TEM results also revealed more severe mitochondrial damage in the LPS-treated, wild-type cells. Recently, mtDNA has been shown to activate the type I interferon (IFN-I) response through the cGAS–STING pathway, further exacerbating inflammation [34]. In addition, we observed mitochondrial accumulation in microglia under inflammatory conditions, with greater accumulation in wild-type cells, which reminds us that TREM2-overexpressing cells may enhance the ability to clear damaged mitochondria. Therefore, we conclude that TREM2 can regulate cellular energy metabolism, reducing the production of ROS induced by inflammation and thereby exerting an anti-inflammatory effect.

Although TREM2 can counteract inflammation exacerbation through metabolic pathways, we observed an increase in reactive oxygen species (ROS) in TREM2-overexpressing cells and a significant accumulation of mitochondria in wild-type cells. We suspect that TREM2 overexpression may affect mitochondrial autophagy in these cells, leading to differences in mitochondrial accumulation between the two groups after LPS treatment. Mitochondrial autophagy is an important means of clearing damaged mitochondria, and can inhibit the inflammation caused by mtDNA release during aging [14]. TREM2 is an upstream molecule of mTOR, which is the most critical autophagy-regulating gene, and mTOR pathway activation inhibits autophagy [5]. Therefore, this may be one of the important mechanisms by which TREM2-overexpressing cells exert an anti-inflammatory effect. After LPS treatment for 4 h, the expression levels of the autophagy-related proteins ATG5 and ATG7 decreased in wild-type cells, indicating a weakened autophagic capacity of microglia during the early stages of inflammation. However, in TREM2-overexpressing cells, the expression levels of ATG5 and ATG7 did not decrease significantly. Additionally, an analysis of the key proteins involved in mitochondrial autophagy, p62 and LC3. LC3 is widely regarded as the gold-standard biomarker for monitoring autophagic activity, owing to its characteristic lipidation-dependent conversion from the cytosolic isoform (LC3-I) to the phosphatidylethanolamine-conjugated membrane-bound isoform (LC3-II) during autophagosome formation. The LC3-II/LC3-I ratio (or LC3 I/II ratio) is commonly used to assess the level of autophagic flux, with a higher ratio indicating increased autophagic activity [35]. Our study revealed that the degree of autophagy was significantly lower in the wild-type cells than in TREM2-overexpressing cells, which maintained a certain level of autophagic activity. This result explains why the TREM2-overexpressing cells presented less mitochondrial accumulation after LPS treatment. Furthermore, previous studies have shown that the autophagy-related protein LC3 can promote the clearance of amyloid proteins by microglia [36], and that maintaining a certain level of autophagic activity may be a key factor in alleviating AD symptoms. This is also one of the reasons why patients with TREM2 gene defects are more susceptible to AD [37,38]. In summary, we believe that TREM2 can alleviate the inflammation exacerbated by mitochondrial damage by maintaining cellular metabolic homeostasis and mitochondrial autophagy activity, thereby suppressing inflammation. 

Although our findings support the role of TREM2 in improving mitochondrial dysfunction for the treatment of neurodegenerative diseases and neuroinflammation, this study has limitations because of its in vitro nature, and cannot fully represent the actual situation in vivo. The lack of in vivo data necessitates further experiments to validate our conclusions. In addition, this study used LPS to generate a neuroinflammatory model; however, this approach may not accurately reflect the true conditions of neuroinflammation. To assess the reliability of our findings, alternative modeling approaches for inflammation should be employed to create neuroinflammatory models that are closer to clinical scenarios. Finally, further research should be conducted to explore other potential molecular mechanisms involving TREM2 and inflammation.

## 5. Conclusions

TREM2 alleviates inflammation by maintaining metabolic homeostasis and mitochondrial autophagy. Overexpressing TREM2 in microglia enhances resistance to inflammation, inhibits LPS-induced metabolic reprogramming, reduces ROS, mitigates mitochondrial damage, and sustains autophagy to clear damaged mitochondria. These findings highlight TREM2’s dual role in suppressing inflammation and protecting mitochondrial function, suggesting its therapeutic potential in neurodegenerative diseases.

## Figures and Tables

**Figure 1 diseases-13-00060-f001:**
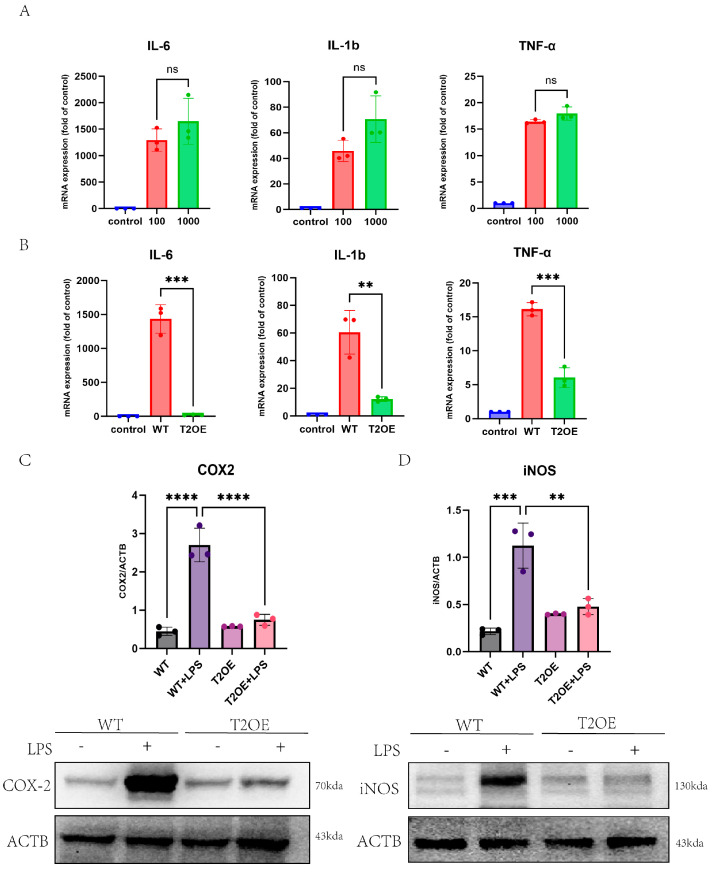
TREM2 inhibits the release of inflammatory cytokines by LPS-treated microglia. (**A**) RT–qPCR was used to analyze the mRNA expression levels of inflammatory cytokines (IL-6, IL-1b, and TNF-α) in wild-type BV-2 cells treated with different concentrations of LPS (100 ng/mL and 1000 ng/mL) for 4 h (*n* = 3). The control group consisted of untreated wild-type cells. (**B**) RT–qPCR was used to analyze the mRNA expression levels of inflammatory cytokines (IL-6, IL-1b, and TNF-α) in TREM2-overexpressing and WT cells after treatment with 1000 ng/mL LPS for 4 h (*n* = 3). The control group consisted of untreated wild-type cells. Western blotting was used to analyze the expression levels of the inflammation-related proteins COX2 (**C**) and iNOS (**D**) in TREM2-overexpressing and wild-type cells treated with or without LPS (1000 ng/mL) for 4 h (*n* = 3). The data are presented as means ± SDs. All results represent three independent experiments. ns: not significant, ** *p* < 0.01, *** *p* < 0.001, and **** *p* < 0.0001.

**Figure 2 diseases-13-00060-f002:**
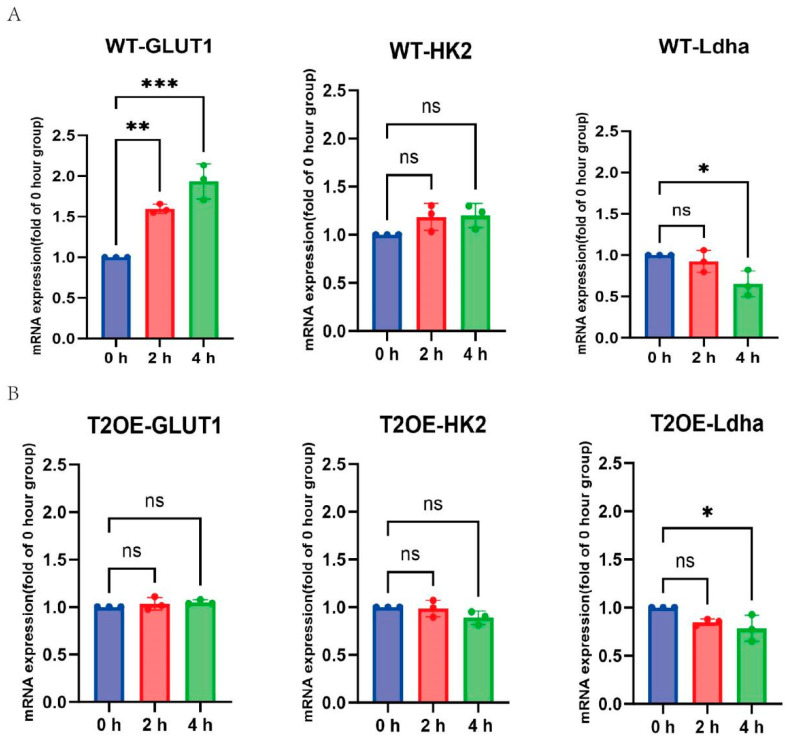
Expression levels of glycolysis-related enzymes in TREM2-overexpressing and wild-type cells after LPS treatment for 0–4 h. (**A**,**B**) RT–qPCR was used to analyze the mRNA expression levels of the glycolysis-related enzymes GLUT1, HK2, and LDHA in TREM2-overexpressing and wild-type cells after treatment with LPS (1000 ng/mL) over time (*x*-axis, in hours) (*n* = 3). The data are presented as means ± SDs. All results represent three independent experiments. ns: not significant, * *p* < 0.05, ** *p* < 0.01, and *** *p* < 0.001.

**Figure 3 diseases-13-00060-f003:**
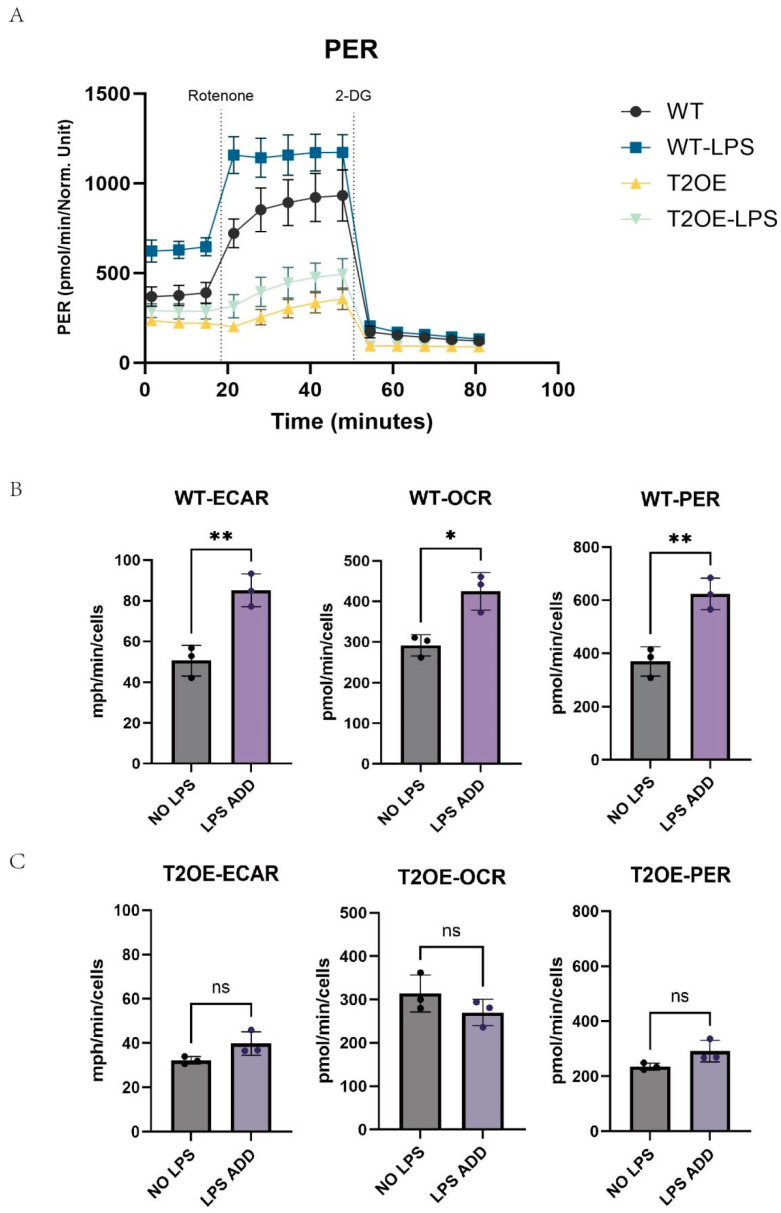
Real-time glycolysis rate in TREM2-overexpressing and wild-type cells after LPS treatment. (**A**) A Seahorse XFp Analyzer was used to assess real-time changes in the proton efflux rate (PER) in cells from each group after treatment with or without LPS (1000 ng/mL) for 4 h (*n* = 3). The dashed line represents the time of drug injection, and the PER from the injection of rotenone to 2-DG represents the maximum glycolysis rate. (**B**,**C**) A Seahorse XFp Analyzer was used to assess the average extracellular acidification rate (ECAR) and oxygen consumption rate (OCR) per minute in TREM2-overexpressing and wild-type cells treated with or without LPS for 4 h (*n* = 3). The data are presented as means ± SDs. All results represent three independent experiments. ns: not significant, * *p* < 0.05 and ** *p* < 0.01.

**Figure 4 diseases-13-00060-f004:**
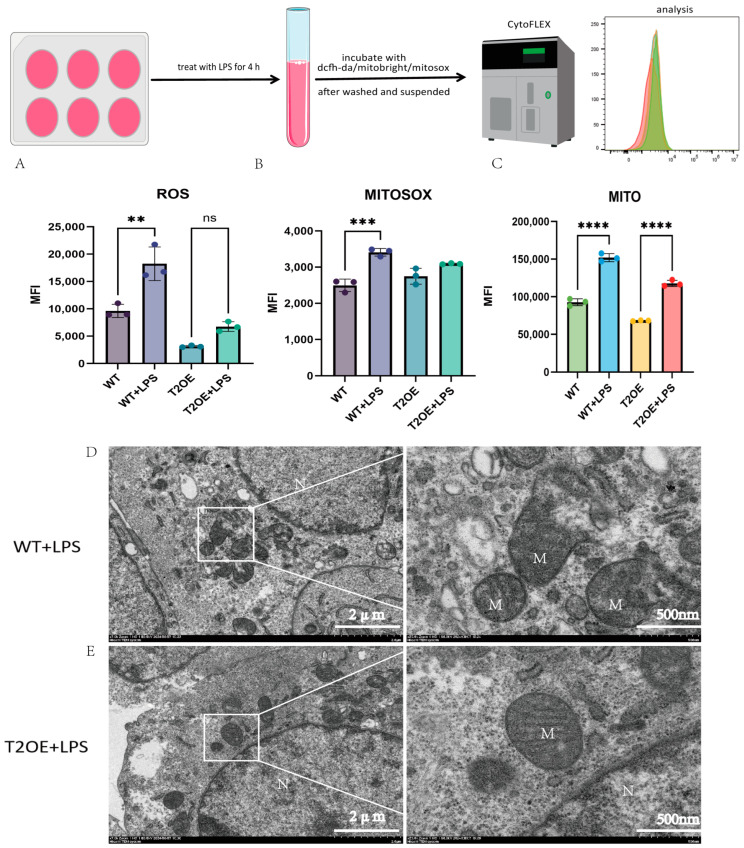
Reactive oxygen species content and mitochondrial status in TREM2-overexpressing and wild-type cells after LPS treatment. The flow cytometry workflow is the top panel. Using different fluorescent dyes (DCFH-DA, MitoSOX, and MitoBright) as probes, flow cytometry was used to detect intracellular ROS (**A**), mitochondrial ROS (**B**), and mitochondrial content (**C**) in cells after LPS (1000 ng/mL) treatment for 4 h (*n* = 3). MFI stands for mean fluorescence intensity. (**D**) Representative transmission electron microscopy images of mitochondria in wild-type and (**E**) TREM2-overexpressing cells after LPS (1000 ng/mL) stimulation for 4 h. M represents mitochondria and N represents the nucleus. Scale bars: 2 μm and 500 nm (white). The images on the right are magnified views of the white boxes in the images on the left. The data are presented as means ± SDs. All results represent three independent experiments. ns: not significant, ** *p* < 0.01, *** *p* < 0.001, and **** *p* < 0.0001.

**Figure 5 diseases-13-00060-f005:**
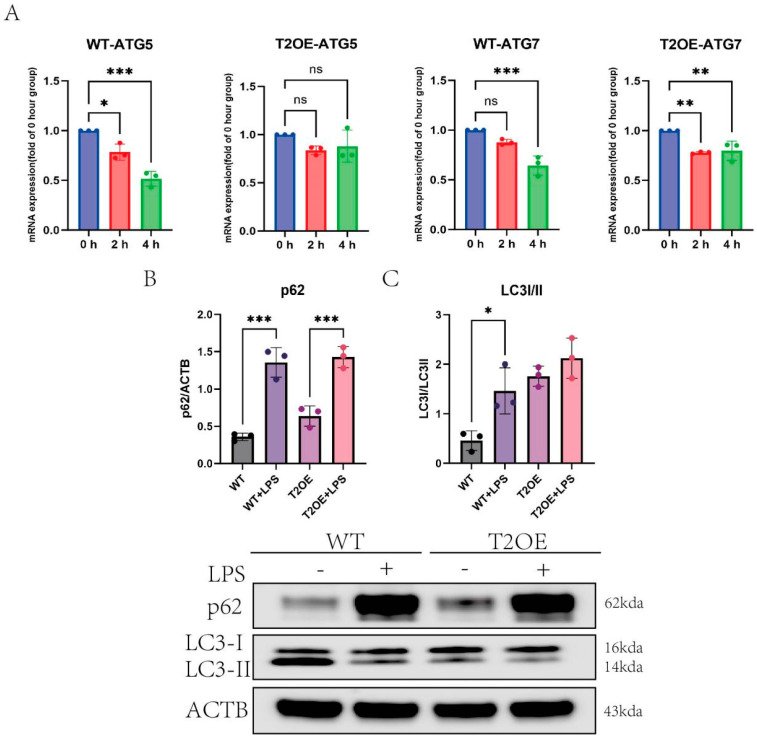
Autophagy in TREM2-overexpressing and wild-type cells following LPS treatment. (**A**) RT–qPCR was used to assess the mRNA expression levels of the autophagy-related proteins ATG5 and ATG7 in TREM2-overexpressing and wild-type cells treated with LPS (1000 ng/mL) over a time course of 0–4 h (*n* = 3). The *x*-axis represents time, with units in hours. (**B**,**C**) Western blotting was conducted to assess the expression levels of the autophagy-related proteins p62 (SQSTM1) and LC3 in TREM2-overexpressing and wild-type cells treated with or without LPS (1000 ng/mL) for 4 h (*n* = 3). The data are presented as means ± SDs. All results represent three independent experiments. Statistical significance is denoted as follows: ns: not significant, * *p* < 0.05, ** *p* < 0.01, and *** *p* < 0.001.

**Figure 6 diseases-13-00060-f006:**
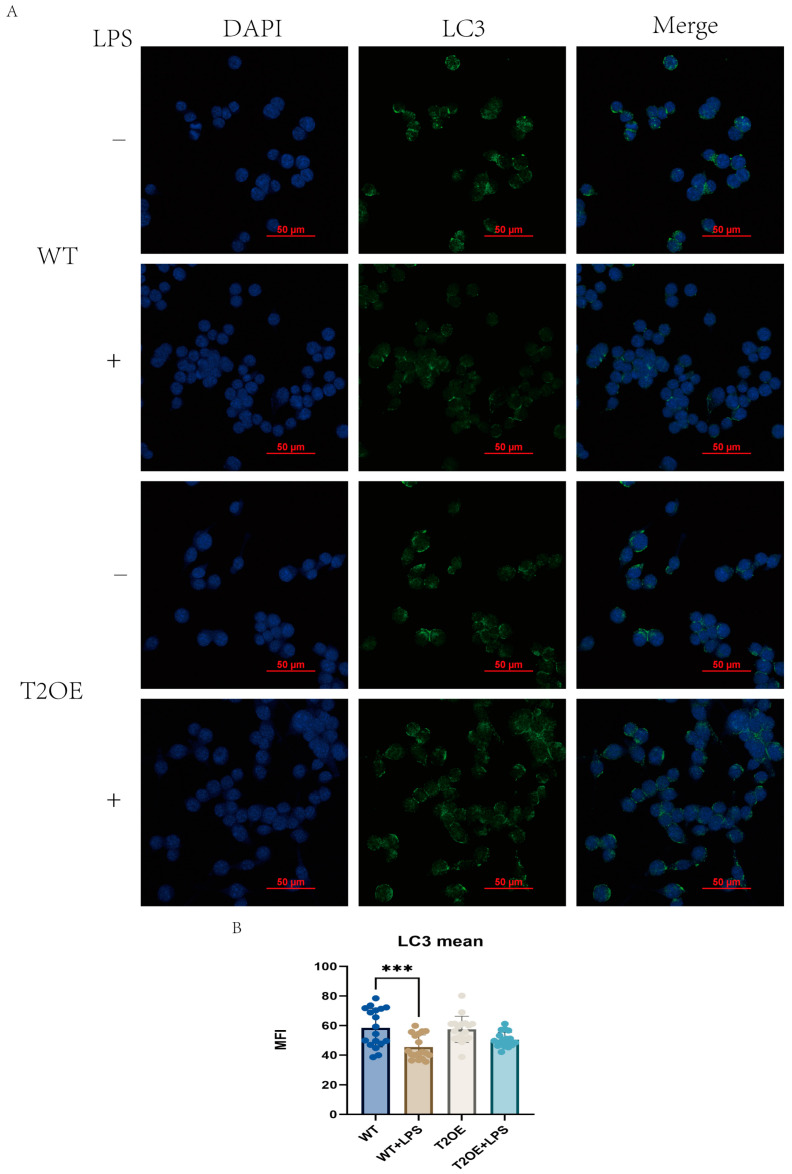
Intracellular LC3 expression in TREM2-overexpressing and wild-type cells after LPS treatment. (**A**) TREM2-overexpressing and wild-type BV-2 cells were treated with LPS (1000 ng/mL) for 4 h, followed by fixation with 4% paraformaldehyde. After the cells were blocked with 5% BSA and stained with an anti-LC3 antibody, nuclei were stained with DAPI and the cells were observed under a laser confocal microscope (*n* = 3, with six randomly selected fields of view counted for each sample). In the immunofluorescence images, the LC3 protein appears as green, while the cell nuclei are blue. Scale bar: 20 μm (indicated in red). (**B**) Mean fluorescence intensity (MFI) of LC3. The results represent three independent experiments, with six different fields of view selected for each experiment to calculate the MFI of LC3. Statistical significance is denoted as follows: *** *p* < 0.001.

**Figure 7 diseases-13-00060-f007:**
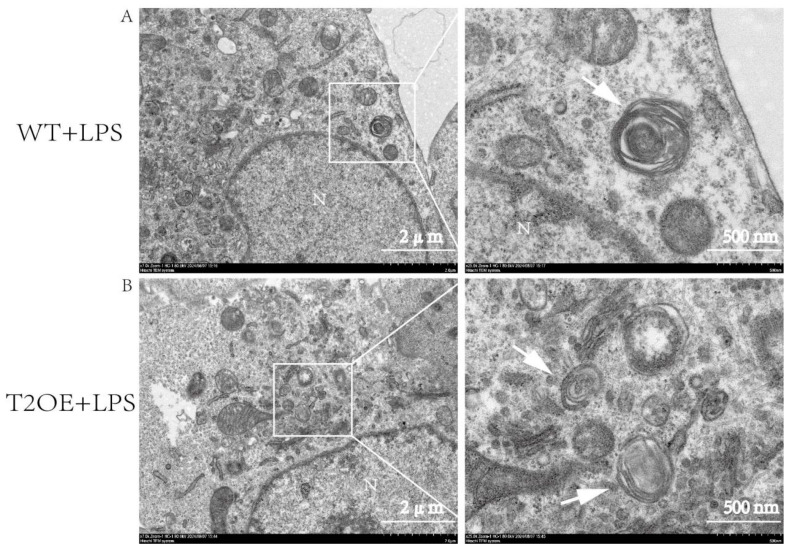
Transmission electron microscopy (TEM) of autophagosomes in TREM2-overexpressing and wild-type cells after LPS treatment. (**A**) Representative TEM images of autophagosomes in wild-type cells. After treatment with LPS (1000 ng/mL) for 4 h, representative transmission electron microscope images of autophagosomes in wild-type cells were captured. N denotes the nucleus. The white arrows indicate autophagosomes. Scale bars: 2 μm and 500 nm (indicated in white). The images on the right are magnified views of the white boxes in the images on the left. (**B**) Representative TEM images of autophagosomes in TREM2-overexpressing cells. After LPS (1000 ng/mL) stimulation for 4 h, representative TEM images of autophagosomes in TREM2-overexpressing cells were obtained. The annotations and scale bars are the same as those described for (**A**).

**Table 1 diseases-13-00060-t001:** Primer sequences for RT–qPCR.

Gene	Forward Primer	Reverse Primer
IL-6	5′ CCAGAAACCGCTATGAAGTTCCT 3′	5′ TGTGTAATTAAGCCTCCGACTTGT 3′
IL-1b	5′ CAGCACATCAACAAGAGCTTCAG 3′	5′ GAGGATGGGCTCTTCTTCAAAGA 3′
TNF-α	5′ GCCTCCCTCTCATCAGTTCTATG 3′	5′ ACCTGGGAGTAGACAAGGTACAA 3′
TREM2	5′ GCGTTCTCCTGAGCAAGTTTCTT 3′	5′ TGTAGTTCTCCTCCCACTCAGAA 3′
GLUT1	5′ TGGTGTCGCTGTTTGTTGTAGAG 3′	5′ AAGATGGCCACGATGCTCAGATA 3′
HK2	5′ TTTGACAGAGAGATCGACATGGG 3′	5′ GGCCTTCTGAATTCCGTCCTTAT 3′
Ldha	5′ GCGTCTCCCTGAAGTCTCTTAAC 3′	5′ CCCGCCTAAGGTTCTTCATTATG 3′
ACTB	5′ GGCTGTATTCCCCTCCATCG 3′	5′ CCAGTTGGTAACAATGCCATGT 3′
ATG5	5′ GTTTGGCTTTGGTTGAAGGAAGA 3′	5′ AATTCGTCCAAACCACACATCTC 3′
ATG7	5′ CGCCAAGATCTCCTACTCCAATC 3′	5′ TGGCATTCACTCCGGGAAATATT 3′

**Table 2 diseases-13-00060-t002:** Primer sequences for mtDNA qPCR.

Gene	Forward Primer	Reverse Primer
NucDNA-Tert	5′ CTAGCTCATGTGTCAAGACCCTCTT 3′	5′ GCCAGCACGTTTCTCTCGTT 3′
MtDNA-Dloop1	5′ AATCTACCATCCTCCGTGAAACC 3′	5′ TCAGTTTAGCTACCCCCAAGTTTAA 3′

## Data Availability

The datasets generated and analyzed during the current study are available from the corresponding author upon reasonable request.

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
