# Peer review of "TREM2 Alleviates Neuroinflammation by Maintaining Cellular Metabolic Homeostasis and Mitophagy Activity During Early Inflammation"

_diseases, 2025, doi:10.3390/diseases13020060_

Round 1
Reviewer 1 Report
Comments and Suggestions for Authors
This is an important study to understand the importance of TREM2 in the cellular metabolic homeostasis. There are a few concerns need to be addressed such as,
- The authors have used only a smaller number of references both in the introduction and discussion. Referencing the prior studies in this field, explaining it and making a case for why the current study is important is crucial. Authors need to improve their manuscript in this regard.
- Either in the methods or in the results, the authors are not clear about how many times they have repeated the experiments for each methodology. There is no n- numbers mentioned anywhere in this manuscript.
- There is no quantifications carried out for the immunofluorescence or the western blot analysis. Do they have a quantification data to present along with the images?
- How many times these cells were grown for each experiments?
- Since it’s a microglia and TREM2 study, did the authors also look in to the Iba1 expressions for microglia? Did the TREM2 over expression also restore structural integrity in microglia?, because its well documented that during inflammation microglial structure changes. Adding Iba1 expression to show the structural integrity will only add more correlation and strength to this study.
- Authors did not mention about the rotenone in the methods, also no mention about what it does when it is introduced to the cells.
- Expand Ldha, HK2, GLUT1 etc., expand shortly what are they either in the methods or in the results.
- In supplemental data, why is the actin shows on a different blot paper. Actin should be developed on the same gel where the other targeted antibodies are developed to make sure the equal loading. Authors should put the data with the actin done along with the targeted proteins on the same blot paper.
Reviewer 2 Report
Comments and Suggestions for Authors
In this research article, the authors examine the relationship between the action of overexpressed TREM2 and metabolic changes in LPS induced inflammation of BV-2 cells. This is an in vitro model of this activity of microglia in early neurodegeneration. They also looked at the connection of the inflammation and autophagy concerning the elimination of damaged mitochondria. This is an interesting point of view and it seems lately that energy metabolism regulation and mitochondrial function are relevant in neurodegeneration. To examine the relationship of these phenomena and autophagy is a new perspective. Their conclusion is that overexpression of TREM2 provides a number of protective effects against inflammation, metabolic energy dysfunction, mitochondrial damage and reduced autophagy.
The experiments are planned and carried out properly, the methods used are modern. The cited literature is up-to-date. The figures are in good quality. The consequences of the results are correct, but some explanation of them is missing.
In 2.10. the authors mention primary microglia cells but otherwise only BV-2 cells appear in the article. In Figure 3, ECAR and PER are lower in the not treated T” overexpressed cells Do the authors have any explanation for this? Similarly: cellular ROS was reduced in the overexpressed cell (Fig. 4.), but there was not real difference with the untreated WT and overexpressed cells in the mitochondrial ROS. Any explanation for this? The mitochondrial content is lower in the untreated T2OE cells compared to the WT cells. What is the ratio of the treated/untreated cells’ mitochondrial content in the two types of cells? Minor remark: GLUT1 is a transporter, not an enzyme (discussion). What could be the line of the events: increased glycolysis then mitochondrial damage in inflammation, or vice versa? Mitochondrial ROS is produced mainly during OXPHOS. What is the reason of the increased cellular ROS? According to Fig. 5, the levels of autophagy proteins elevated in the overexpressed cells compared to the WT. What is the reason and consequence of this? Could you explain the significance of the LC3 I/II ratio, and why was in elevated in the overexpressed cells?
Round 2
Reviewer 1 Report
Comments and Suggestions for Authors
The authors have significantly improved the manuscript on revision.